# Enhanced Protection of Biological Membranes during Lipid Peroxidation: Study of the Interactions between Flavonoid Loaded Mesoporous Silica Nanoparticles and Model Cell Membranes

**DOI:** 10.3390/ijms20112709

**Published:** 2019-06-01

**Authors:** Lucija Mandić, Anja Sadžak, Vida Strasser, Goran Baranović, Darija Domazet Jurašin, Maja Dutour Sikirić, Suzana Šegota

**Affiliations:** 1Ruđer Bošković Institute, Division of Physical Chemistry, 10000 Zagreb, Croatia; Lucija.Mandic@irb.hr (L.M.); Anja.Sadzak@irb.hr (A.S.); Vida.Cadez@irb.hr (V.S.); Darija.Jurasin@irb.hr (D.D.J.); sikiric@irb.hr (M.D.S.); 2Ruđer Bošković Institute, Division of Organic Chemistry and Biochemistry, 10000 Zagreb, Croatia; Goran.Baranovic@irb.hr

**Keywords:** lipid peroxidation, membrane elasticity, mesoporous silica nanoparticles, myricetin, myricitrin, nanomechanics, protective effects of flavonoids, quercetin

## Abstract

Flavonoids, polyphenols with anti-oxidative activity have high potential as novel therapeutics for neurodegenerative disease, but their applicability is rendered by their poor water solubility and chemical instability under physiological conditions. In this study, this is overcome by delivering flavonoids to model cell membranes (unsaturated DOPC) using prepared and characterized biodegradable mesoporous silica nanoparticles, MSNs. Quercetin, myricetin and myricitrin have been investigated in order to determine the relationship between flavonoid structure and protective activity towards oxidative stress, i.e., lipid peroxidation induced by the addition of hydrogen peroxide and/or Cu^2+^ ions. Among investigated flavonoids, quercetin showed the most enhanced and prolonged protective anti-oxidative activity. The nanomechanical (Young modulus) measurement of the MSNs treated DOPC membranes during lipid peroxidation confirmed attenuated membrane damage. By applying a combination of experimental techniques (atomic force microscopy—AFM, force spectroscopy, electrophoretic light scattering—ES and dynamic light scattering—DLS), this work generated detailed knowledge about the effects of flavonoid loaded MSNs on the elasticity of model membranes, especially under oxidative stress conditions. Results from this study will pave the way towards the development of innovative and improved markers for oxidative stress-associated neurological disorders. In addition, the obtained could be extended to designing effective delivery systems of other high potential bioactive molecules with an aim to improve human health in general.

## 1. Introduction

Oxidative stress is one of the major causes of neuronal death in a variety of neurodegenerative diseases [1]. It occurs when cellular antioxidant defense is insufficient to keep the levels of reactive oxygen species (ROS) below a toxic threshold. Antioxidant defense engaged in maintenance of redox homeostasis is provided by various biological antioxidants such as reduced glutathione (GSH) and by diverse antioxidant enzymes. Among different biological molecules, polyunsaturated fatty acids (PUFAs) abundant in neuronal membranes are highly prone to ROS-induced lipid peroxidation, a chain reaction of free radical formation in the lipid parts of cellular membranes. This is particularly important in relation to the brain, because the brain is highly vulnerable to oxidative damage [2,3]. It is considered that a decrease of ROS generated by antioxidants could be an effective therapeutic strategy in neuroprotection. High potential of the flavonoids to regain redox homeostasis and prevent or delay neuronal oxidative injury is strongly emphasized in recent years [4]. 

Flavonoids are a broad class of polyphenolic biomolecules, with numerous hydroxyl groups, found in a variety of fruits and vegetables. They exert different biological activities such as, anticarcinogenic [5], antiinflammatory [6] and antibacterial activity [7]. Presumably they possess remarkable therapeutic potential in preventing the onset and progression of Alzheimer’s disease and in promoting cognitive performance [8]. They might facilitate a protective or preventive effect in model systems for studying Alzheimer’s disease [9]. However, the mechanisms involved in antioxidant effects of flavonoids have not yet been fully elucidated. In addition, the use of flavonoids has been limited due to their poor water solubility, i.e., high hydrophobicity, and chemical instability under physiological conditions [10].

A promising way to deliver poorly soluble bioactive molecules is their incorporation into nanoparticles (NPs) [11,12,13]. Among different NPs, biodegradable NPs are gaining increased attention for their ability to serve as a viable nanocarriers for site-specific delivery of biomolecules in the body and offer enhanced biocompatibility and convenient release profiles for a number of drugs, vaccines and biomolecules [14]. Up to now for flavonoid delivery, different organic (liposomes, dendrimers, polymer NPs and lipid NPs [15,16]) and inorganic (gold NPs, TiO_2_ NPs and Fe_3_O_4_ [17,18,19]) NPs were used. Lately biodegradable mesoporous nanoparticles MNPs emerge as the ones having ideal properties for designing a nano-particulate delivery system: Effectively controlled particle size and surface chemistry; enhanced permeation, flexibility, solubility and release of therapeutically active agents in order to attain the target and specific activity at a predetermined rate and time [20,21]. Due to their large active surface area and high pore volume, they are able to host diverse molecules and as a result have the highest drug-loading efficiency [13,20,22]. Besides, they can provide excellent physico-chemical protection from flavonoid degradation at physiological conditions, particularly during endogenic enzymatic activities [12,20]. Mesoporous silica nanoparticles (MSNs) presents promising flavonoids nanocarriers due to their high loading efficacy, sustained release properties and subcellular size. They are widely used for the controlled delivery of drugs and proteins [11,12,20,23,24,25]. In order to confirm the usefulness of this approach, we will also investigate interactions between MSNs and membranes since they are crucially important both for cell uptake and nanotoxicity [26].

Within this study, flavonoids from the subgroups of flavonols, quercetin [27,28], myricetin [29] and myricitrin [30] have been selected. Literature data show that upon therapeutic intervention with quercetin, significant neuroprotection as well as neuronal recovery can be achieved [27,28]. Myricetin is a natural flavonol from fruits, vegetables, tea, berries, red wine and medical plants. Myricetin exhibits antioxidative, anticancerogenic and antiinflammatory effects with strong scavenging activity [29] and is able to protect a variety of cells from in vitro and in vivo injury [6,29,31]. However, the antioxidant ability of myricetin in vitro is less apparent in vivo due in part to a low solubility and relatively poor bioavailability [20]. Myricitrin reportedly possesses effective antioxidative effect with strong free radical scavenging activity [30]. Moreover, the antiinflammatory potential [32], antinociceptive effect [33] and neuroprotective action [33] findings suggest that myricitrin exhibits a significant heteroprotective role. All three flavonoids contain planar moiety but differ in the degree of monosaccharide unit substitution. Therefore, choice of flavonoids enables determination of a relationship between flavonoid structure and protective activity towards oxidative stress [34].

Flavonoids are capable to interact with biological and cell membranes and penetrate more or less deep into their hydrophobic or interphase sites, particularly compartments known as lipid rafts, depending on the hydrophilicity and/or hydrophobicity of selected flavonoid. Consequently, membrane hosted flavonoids influence on its thermotropic (phase transition temperature) [35,36,37], biophysical (microviscosity, transmembrane potential) [3,36], and nanomechanical properties (elasticity, e.g., fluidity and permeability) [38] altering the membrane constituents arrangement responsible for cell signal transduction, the regulation of the metabolism and biological activity [39]. These interactions are essentially responsible for the development of novel drugs and at least enabled the deep insight of their therapeutic potentials. The complex cell membrane structure along with highly dynamic membrane processes, particularly interactions with drugs and drug delivery systems, are very difficult to investigate from the biophysical point of view. Therefore, simplified model membrane systems (liposomes, supported lipid bilayers (SLBs) and lipid monolayers) have been developed that are the subject of numerous long-term researches. As the model membrane constituent, phospholipid, particularly unsaturated dioleoyl-phosphatdylcholine (DOPC) has been investigated [40]. The advantage of using such model lipid membranes without other membrane components lies in their avoidance of the interference that enables the focus on the process of their mutual interaction with lipid membranes [41].

In numerous studies the interactions between membranes and flavonoids have been investigated, but the mode of their action still remains partially understandable. Besides functional proteins (enzymes) flavonoids react with lipid bilayers and influence membrane properties. The hydrophilic flavonoids form the hydrogen bonds between flavonoid molecules and the polar membrane interface [42] inducing the membrane rigidification. On the other hand, the more hydrophobic flavonoids showed marked affinity for the membrane interior and therefore caused significant membrane permeability. Consequently, both polar and nonpolar forces were shown to have a significant impact on the flavonoid-membrane interactions [43]. Induced rigidification effect is presumed to hamper the radical diffusion and decreases the kinetics of radical reactions within the membrane environment, resulting in the inhibition of the lipid peroxidation process [44]. However, the modification (increase or decrease) of membrane fluidity is undoubtedly responsible for the antioxidative effects of flavonoids, or drugs, in general.

In this study we achieved three goals: (i) Increase of flavonoid loading efficiency by mesoporous silica nanoparticles (MSNs), as compared to so far used organic or inorganic NPs; (ii) protection of flavonoids from chemical degradation under physiological conditions and (iii) sustained flavonoid release enabling their interaction with the model cell membrane.

The innovation provided by this study is the measurement of the membrane structural reorganization induced by peroxidation/copper ions by combining atomic force microscopy (AFM) imaging as well as non-imaging data. Among investigated flavonoids, quercetin, incorporated in MSNs, showed the most enhanced and prolonged protective anti-oxidative activity. The nanomechanical (Young modulus) measurement of the MSNs treated DOPC membranes during lipid peroxidation confirmed an attenuated lipid peroxidation. By applying a combination of experimental techniques that are not fully exploited until now in the field of molecular biotechnology (atomic force microscopy (AFM), force spectroscopy (FS) and dynamic/electrophoretic light scattering (DLS/ELS)), this study ultimately generated detailed knowledge about the effects of the structure and hydrophobicity of flavonoids loaded in MSNs on model lipid membranes under conditions of oxidative stress. Specific information about how the structural and nanomechanical properties of model membranes change as a valuable indicator has been provided. The nanomechanics (elasticity) and surface topography (roughness) of model lipid membranes that result from oxidative damage have not yet been quantified at the nanoscale.

## 2. Results and Discussion

### 2.1. Preparation of Mesoporous Silica-PEG Nanoparticles (MSNs)

The scheme of the MSNs synthesis is presented on Figure 1.

Modification of the amine functionalized MSNs (A-MSNs) was achieved by mixing it with mPEG-SG to form covalent bonds between the amine groups on the outer surface of amine and the succinimidyl groups of mPEG-SG as showed on Figure 1. The characterization of propyl-amine MSNs used in experiments were characterized by X-ray powder diffraction (XRPD), Brunauer–Emmet–Teller (BET) analysis, atomic force microscopy (AFM), field emission scanning electron microscope (FE-SEM) and electrophoretic (zeta potential) and dynamic light scattering (DLS) measurements. Data are summarized in Table 1 and Figure 2.

The diffraction pattern of the investigated powdered MSNs is not characterized by sharp diffraction lines but shows a typical halo in the 2θ = 20°–30° region, which proves the amorphicity of the sample, i.e., indicates short-range atomic ordering between the Si and O at the prepared MSNs. The distribution of diameters of MSNs obtained on the powder sample (Figure 2D) using an FE-SEM micrograph amounted to 326 ± 137 nm (Figure 2D, *n* = 57), which is in agreement with the AFM imaging results where the average diameter of the observed MSN (Figure 2B,C) had been determined to be around 300 nm, while dispersed MSNs in water tended to aggregate resulting in a high increase of the diameter with the hydrodynamic diameter of aggregates obtained by dynamic light scattering (DLS; *d*_H_ = 913 ± 180 nm). The measured value of the zeta potential (*ζ* = +26 ± 2 mV) confirms that the nanoparticles are positively charged and are stable in the aqueous medium. Unmodified MSNs had a negative zeta potential, (*ζ* = −50 mV) in a wide pH range reflecting the large surface charge due to deprotonated Si–OH groups. Conversely, functionalized MSN with –NH_2_, –CH_3_ and –OH groups may have a positive, neutral or negative charge, depending on the pH medium [45]. Therefore, in our case, the positive zeta potential comes from the propylamine of MSNs functionality.

We further performed the experiments to stabilize MSNs. PEG is hydrophilic, biocompatible and non-toxic and can therefore delay hydrolysis and enzymolysis [46]. PEG prevents protein adsorption (opsonization) on the surface of NPs and decreases non-specific intake into the reticuloendothelial system [47]. The amount of PEG that can be incorporated into the liposome lipid bilayer decreases with an increase in molecular weight of PEG [48] of 15 mol% in PEG 120 to 5–7 mol% for PEG 2000 and PEG 5000. Above 7.5 mol% for PEG 1900 comes liposome dissolution [49]. The best conditions for avoiding adsorption of biomolecules to surface MSNs are with long PEG chains and high surface density [50].

Part of the samples of PEG-coated MSNs was used for the analysis and the characterization. The results from AFM, FTIR spectroscopy and Field Emission Scanning Electron Microscopy(FE‒SEM ) are depicted on Figure 3.

AFM 2D and 3D height images (Figure 3A,C) and FF-SEM image (Figure 3E) show morphology of mesoporous MSNs coated with PEG (M_w_ = 5000). The PEG coating was confirmed on the phase image (Figure 3B, white color on the surface of the MSN). Fine PEG coating was confirmed by FTIR spectroscopy (Figure 3D) showing MSNs spectra with different mass ratios of PEG. The peak at 2910 cm^−1^ was assigned to methylene stretching (CH_2_) in PEG, a ribbon stretching Si–O (1630 cm^−1^) and Si–C stretching (876 cm^−1^). Other peaks that do not belong to pure MSN are at 1710, 1507, 1466 and 1352 cm^-1^. The last two are due to methylene vibrations of angular vibrations. The average hydrodynamic diameter of MSNs dispersed in water was *d*_H_ = 932 ± 91 nm (Table 2) indicating MSNs aggregation process within water dispersion of MSNs. Although the results of DLS showed that MSNs were to some extent aggregated, they remained stable for a prolonged time period with a zeta potential of 27 ± 1 mV (Table 2). Applied methods in the characterization of MSNs, namely FF-SEM, AFM, FTIR spectroscopy and dynamic and zeta potential measurements, confirmed the mesoporosity of the prepared nanoparticles, particularly those coated with PEG (Mw 5000; in weight ratio w(MSNs):w(PEG) = 1:5) as the most stabilized.

### 2.2. Loading of Flavonoids into MSNs

The experiments were performed to determine the loading efficiency (LE) at one flavonoid MSNs weight ratio, 3:1. The LE for quercetin, myricetin and myricitrin was 27% ± 9%, 4% ± 2%, and 8.6% ± 0.6%, respectively. Figure 4A,D,G showed the morphology of the MSNs loaded with quercetin, myricetin and myricitrin, respectively. They kept the size and the morphology of those before loading. After the loading of MSNs with flavonoids, flavonoid loaded MSNs were washed with EtOH to remove the unadsorbed flavonoids, the part of them stayed adsorbed on the surface of the MSNs and thus induced the aggregation as it is observed on the AFM phase Figure 4B,E,F. The BET analyses confirmed the loading of flavonoids by a decrease in the free specific surface areas, pore volumes and pore sizes (Table 3). Thus the specific surface area decreased from 693.78 m^2^ g^−1^ for empty MSNs to 544.58 m^2^ g^−1^, 546.01 m^2^ g^−1^ and 562.78 m^2^ g^−1^ for MSNs loaded with quercetin, myricetin and myricitrin, respectively. The equal trend was observed for the pore volume and pore size of MSNs loaded with flavonoids. Aggregates made of a few number of single MSNs (Figure 4) has an active surface that is somewhat smaller than the sum of surfaces of all individual MSNs. The MSNs, if composed of almost rigid spheres, can have only open pores, i.e., anything that enters will eventually find its way out. As determined experimentally the pore sizes were between 4.82 nm and 3.12 nm. Knowing that the characteristic dimensions of a flavonoid molecule are all around 1 nm (its length is slightly below 2 nm, besides the myricitrin molecule as the glycone molecule is the largest among them), flavonoid molecules that could be entrapped within the pores due to their hydrophobicity and van der Waals forces correspond to their sizes. Thus, for the most hydrophilic flavonoids used, the myricetin shows the highest hydrophobicity, and should have the smallest LE. FTIR spectroscopy measurements (Figure 4C,F,I) confirmed also a successful loading of flavonoids within MSNs. A detailed analysis of the spectra (outside the scope of this research) confirms the presence and structural integrity of flavonoid molecules within the silica pores. Moreover, changes in the position of flavonoid vibrational modes between 1400 and 1600 cm^−1^, more precisely 1503, 1458 and 1379 cm^−1^ and 1349 (w) cm^−1^ and 1461 (w) cm^−1^ for quercetin, myricetin and myricitrin loaded MSNs, respectively, i.e., ν(C=O), aromatic ν(C=C) and ν(C–O/δO–H) combination modes, indicate hydrogen bonding interactions from the silica surface groups to the carbonyl of flavonoids. For quercetin loaded MSNs, a band at 1153 cm^−1^ was attributable to the C–O stretching in the aryl ether ring and the C–O stretching in phenol, respectively [51]. Bands at 2930 (sh), 2857 cm^−1^ were due to CH stretching of PEG. The band at 1379 cm^−1^ belongs to myricetin (Figure 4F), while in the spectrum of myricitrin loaded MSNs are present bands 2930, 1507, 1461 and 1346 (w) cm^−1^. (Figure 4I). Band at 1346 cm^−1^ belongs present myricitrin.

### 2.3. Release of Flavonoids from MSNs

The cumulative release profiles for the selected flavonoids (Figure 5A,B) are practically unchanged, i.e., essentially not dependent on the flavonoid structure. In an early stage of the release, a burst was observed lasting firstly for eight hours. It is thus supposed that around 1.5%, 0.4% and 0.5% of loaded quercetin, myricetin and myricitrin, respectively, are situated at the outer surface of MSNs. After 24 h the release of all three flavonoids changed its character pointing to a different way of the releasing molecules that were adsorbed. In other words, the release reached a slight plateau where it was more or less constant. The most flavonoids molecules were thus entrapped into the mesoporous cavities of MSNs. The mechanism of flavonoid release was not specific as already confirmed for the LE meaning that the selected flavonoids were adsorbed to the silica surface via deprotonated catechol groups. After 170 h only 1.6%, 0.45% and 0.55% of quercetin, myricetin and myricitrin, respectively, had been released. While within a MSNs flavonoid molecule it can last longer, as soon as it is released, its half-life in the medium is only a few days and therefore two competitive processes have to be simultaneously treated during the drug release. The average half-life of quercetin absorbed in a human organism is 3.5 h [52]. The half-life of myricetin at pH 5 PBS buffer is eight days, while at pH 8 it is only 0.1 h [53]. The biodegradation half-life of myricitrin is 26 days [54]. The fact that the flavonoids loaded into MSNs remained stable during a prolonged period of time thus presents a good improvement. A similar improvement in the sense of a prolonged release has been found also in the quercetin release from poly-lactide NPs showing almost 60% of the released quercetin after four days. When quercetin was released from dextran coated NPs the dissolution rate was linear with time with the slope changing after 10 days [18]. The maximum values (70% and 80%, respectively) were achieved after 15 days.

### 2.4. Interaction of MSNs with Model Cell Membranes

#### The Protective Role of Released Flavonoids into Model Membranes and Their Protective Role during H_2_O_2_ Induced Lipid Peroxidation

Our first issue was to answer to the question whether the flavonoids in respect to their difference in hydrophobicity, or structure, and molar ratio towards lipids are able to insert lateral homogeneously in the bilayer and whether the inserted flavonoids protect the model lipid membrane towards the lipid peroxidation process. To get the answer to this question, we performed zeta (ELS), AFM and force spectroscopy measurements.

The insertion of the flavonoid (quercetin, myricetin and myricitrin) has been checked by zeta potential measuring of the prepared DOPC liposomes, without and with inserted flavonoids. The obtained results are presented in Table 4 as the average value ± standard deviation of five independent measurements for each sample. Recalling the zeta potential of the pure DOPC liposomes (*ζ* = −4.2 mV) [55] and comparing it with zeta potentials of DOPC liposomes with flavonoid loaded MSNs, we concluded that in all cases, flavonoids released from MSNs during incubation with DOPC liposome at 25 °C, were inserted into the DOPC bilayer as denoted by the shift of the zeta potential to negative values (from −6.1 ± 1.1 mV for pure DOPC to −14.4 ± 4.7 mV, −6.7 ± 1.3 mV and −11.8 ± 3.2 mV for quercetin, myricetin and myricitrin, respectively). The similar behavior has been observed in the study of interaction of glucone hysperidin and aglicone hesperitin with a DMPC bilayer [38], in that has been established that the insertion of flavonoids followed by changes in zeta potential values, that turns down near the saturation of the bilayer with flavonoids. However, the valuable information could be thrown up from the results shown above and be sufficient for following the conclusion. First, as shown in this study for DOPC in the PBS buffer solution, the electrophoretic mobility of DMPC liposomes was low, but not negligible. Our result obtained in the flavonoid/DOPC study (*ζ* = −6.21 ± 1.1 mV at 25 °C for the fluid phase) is in agreement with already reported (ζ = −4.2 mV) [55] confirming the proper sample preparation and the reproducibility of the results. The net negative zeta potential value in the PBS buffer solution confirmed the binding of the ions present in the buffer solution at *I* = 0.15 M. Second, recalling of the p*K*_a_ = 5.87 and 8.48 [56], p*K*_a_ = 6.33 [57] and p*K*_a_ = 5.23 [58] for quercetin, myricetin and myricitrin, respectively, the anion species of quercetin, myricetin and myricitrin (deprotonated species) are predominant at pH 7.4. The shift in zeta potential towards negative values might be ascribed only to the addition of flavonoids. This is taken as a strong indication that the increase in the surface negative charge is due to flavonoid insertion into the DOPC liposomes. After addition of H_2_O_2_, zeta potential shifts towards positive values have been observed from −14.4 ± 4.7 mV to −8.1± 1.4 mV and from −11.8 ± 3.2 mV to −6.2 ± 2 mV for quercetin and myricitrin, respectively. In contrast, the shifts towards a more negative zeta potential had been observed from −6.1 ± 1.1 mV to −17.8 ± 6 mV and from −6.7 ± 1.3 mV to −10.4 ± 2.2 mV for pure DOPC and myricetin, respectively. That could indicate that the mechanism of membrane protection occurred differently for structurally different flavonoids in respect to their locations within the membrane. It is important to highlight that the shift in zeta potential values were suppressed in the presence of the flavonoids (Δ*ζ* = +6.3 ± 6.1 mV, Δ*ζ* = −3.7 ± 3.5 mV and Δ*ζ* = +5.5 ± 5.2 mV) for quercetin, myricetin and myricitrin, respectively, in comparison to pure DOPC liposomes (Δ*ζ* = −11.7 ± 7.6 mV). These measurements in some extend the protective role of used flavonoids under induced oxidative stress.

Therefore, our next issue was to answer the question whether the flavonoids in respect to their difference in hydrophobicity and structure towards lipids are able to insert lateral homogeneously in the bilayer. To get an answer to this question, we performed AFM measurements. The topography of SLB with an inserted flavonoid before and after the addition of H_2_O_2_ and Cu^2+^ ions in PBS obtained for different samples at 25 °C is presented in Figure 6 and Appendix A. By adjusting the pH = 7.4, the time and conditions of imaging, the formation of SLB was optimized and therefore the changes in topography of the SLB correspond only to the differences between the samples. The cross sections profiles (Appendix A) as well as the thickness determination of the bilayer from the force curves (Appendix A) show the profiles corresponded to the single supported bilayer covering the mica support.

The homogeneous scabrous SLBs without ruptures are clearly seen irrespective of inserted flavonoids. The roughness of the SLB for all examined protrusions from the unperturbed SLB surface are corrected for the convolution effect of the tip [59] and presented in Table 5. The roughness of the different domains of the SLBs has been calculated by four random average root mean square (*R*_a_) values on area 2 × 2 µm^2^.

The average roughness of control DOPC SLB amounts Ra = 0.08 ± 0.01 nm indicating very smooth SLB, while the moderately rougher surface of SLBs exposed MSNs containing quercetin, myricetin and myricitin showed roughness *R*_a_ = 0.11 ± 0.05 nm, 0.12 ± 0.06 nm and 0.19 ± 0.09 nm, respectively. This difference in the roughness in comparison to the control DOPC SLB (Δ*R*_a_ = 0.03, 0.04 and 0.14 nm for quercetin, myricetin and myricitrin, respectively) could be explained only by an increased surface density of the observed protrusions on the investigated area (25 µm^2^ caused only by insertion of flavonoids. Recalling the hydrophobicity/hydrophilicity, i.e., partition coefficients of examined flavonoids (log *p* = 1.86, 1.75 and 0.25 for quercetin, myricetin and myricitrin, respectively) [60], significant higher roughness was observed for more hydrophilic flavonoid glucone myricitrin in comparison to the other. Since all SLB were performed under the same experimental conditions, the observed difference in the roughness value seems to be a good indicator for the incorporation of the released flavonoids from the MSNs into the DOPC liposomes during 48 h of exposure.

Now we turn back to the question of the effects of the flavonoid on the nanomechanical properties of DOPC SLB. The effect of flavonoids on the nanomechanics was further investigated in detail by force spectroscopy. The elasticity maps (Figure 7) showed values of the elasticity that could be attributed to the fluid phase of DOPC. A distinct decrease of the Young modulus was observed from *E* = 63.7 ± 5.2 MPa (for control DOPC SLB) to *E* = 40.6 ± 2.7 MPa, 31.4 ± 2.9 MPa and 37.6 ± 4.8 MPa (for quercetin, myricetin and myricitrin, respectively, exposed pure DOPC). The observed decrease in the Young moduli of SLB indicated an insertion of exposed flavonoids hosted in MSNs that have been released during the incubation time. However, the insertion of the flavonoids occurred without any impact on SLB morphology and homogeneity, i.e., reduced membrane stiffness (or absence of elasticity) has not been enough to disorganize or destabilize the whole SLB structure by pore formation. These behaviors could be attributed to the decreased lipid lateral interactions in PBS provoked by the insertion of the flavonoids. The observable increased fluidities of the DOPC SLB with inserted flavonoids with respect to the pure DOPC SLB amounted to Δ*E* = −23.1 ± 7.9 MPa, 32.3 ± 8.1 MPa and 26.1 ± 10.0 MPa for quercetin, myricetin and myricitrin, respectively. Without taking in consideration the concentration of released flavonoids during 48 h, at the same experimental conditions, the highest effect on the elasticity of DOPC SLB showed hydrophobic aglycone myricetin indicating that the affinity or activity of myricetin to DOPC SLB appeared higher than the other two flavonoids. By analogy recently reported studies on SLBs with aglycone hesperetin and glycone hesperidin [36,61], the permeation is not expected to be equal for glycone myricitrin and aglycone myricitrin. This indicates that bilayer disordering was caused by the flavonoid presence as well as the insertion of the flavonoids is in agreement with corresponding partition coefficients. The presence of flavonoids, particularly hydrophilic myricitrin near the lipid phosphate groups modified the orientation of the bilayer dipoles and consequently the mutual interaction between the phospholipids. On the other side, the inclusion of flavonoids, at low concentrations conditions showed a moderate effect on lipid rearrangement, causing the expansion of the nonpolar domains within the bilayer and thus the increase in the membrane elasticity and bilayer thickness.

Our next issue was to answer the question whether the inserted flavonoids protect the model lipid membrane towards an induced lipid peroxidation process by the addition of H_2_O_2_ and Cu^2+^ ions. To get the answer to this question, we performed further AFM and force spectroscopy measurements and analysis.

The topography of SLB with an inserted flavonoid after the addition of H_2_O_2_ and Cu^2+^ ions in PBS obtained for different samples at 25 °C is presented in Figure 6E,H,K and Figure 6C,F,I,L, while the cross section profiles show the profiles corresponding to the single supported bilayer covering the mica support (Appendix A). Opposite to pure DOPC SLB, the inhomogeneous scabrous SLBs with protrusions are clearly seen irrespective of inserted flavonoids. The average roughness of pure DOPC SLB exposed to H_2_O_2_ and H_2_O_2_ and Cu^2+^ ions showing the quite rough surfaces increased from *R*_a_ = 0.08 ± 0.01 nm to *R*_a_ = 0.33 ± 0.05 nm and *R*_a_ = 0.84 ± 0.02 nm, respectively, indicating a significantly disrupted surface during the lipid peroxidation process. The increases of the roughness have been shown to be Δ*R*_a_ = +0.25 ± 0.06 nm and Δ*R*_a_ = 0.76 ± 0.01 nm, for the addition of H_2_O_2_ and H_2_O_2_ and Cu^2+^ ions, respectively. The same, but reduced effect of the induced lipid peroxidation process was observed in all SLBs with inserted flavonoids, indicating the protective role of flavonoids towards lipid peroxidation. The damage of the SLB induced by the H_2_O_2_ addition was reduced from Δ*R*_a_ = +0.25 ± 0.06 nm to Δ*R*_a_ = 0.07 ± 0.11 nm, 0.10 ± 0.08 nm and 0.11 ± 0.13 nm for treated SLBs by quercetin, myricetin and myricitrin, respectively. The protective role of the present flavonoids is also observable in the treatment of SLBs with H_2_O_2_ + Cu^2+^, but the protection is much more suppressed indicating a different mechanism of the process of lipid peroxidation (Table 5).

Mutually comparing the protection of SLBs by insertion of different flavonoids, it could be concluded, that the quercetin had the highest protective activity measured by the roughness parameter. The structural reorganization of the lipid molecules and the SLB damage during and after lipid peroxidation were reflected not only in the bilayer thickness but also in the Young moduli values obtained during force spectroscopy measurements. By analyzing force distance curves accurate values for the bilayer height were thus obtained, 7.0 ± 0.3 nm, 7.2 ± 0.4 nm, 7.2 ± 0.6 and 7.3 ± 0.2 nm for pure DOPC, quercetin, myricetin and myricitrin loaded SLB, respectively.

Imaging was carefully repeated a few times to prevent any resonant frequency change due to contamination. The results summarized in Table 5 and Appendix A show that the observed thickness of DOPC SLB is in agreement with the already reported by Atwood [40]. The bilayer thickness increasing due to the effect of flavonoid insertion additionally confirmed their effective incorporation within the bilayer, as has been observed by zeta potential measurements of liposomes. By H_2_O_2_ inducing the lipid peroxidation, the SLBs were thinner than those before lipid peroxidation with an effective thicknesses of 6.9 ± 0.2 nm, 7.0 ± 0.3 nm, 7.0 ± 0.2 nm and 7.1 ± 0.3 nm for pure DOPC, quercetin, myricetin and myricitrin loaded SLB, respectively. By lipid peroxidation induced by the addition of both H_2_O_2_ and Cu^2+^ the SLBs showed an effective thicknesses of 6.7 ± 0.5 nm, 6.8 ± 0.5 nm, 6.9 ± 0.5 nm and 7.0 ± 0.4 nm for pure DOPC, quercetin, myricetin and myricitrin loaded SLB, respectively. The observed decrease in the bilayer thickness regarding the influence of the process of lipid peroxidation on the overall membrane structure indicates lipid molecule restructuring within the bilayer. This process could lead to some extent membrane damage that is reflected again in the membrane elasticity increases.

The elasticity maps shown on Figure 7B,E,H,K present fluid DOPC SLBs after the addition of H_2_O_2_, while Figure 7C,F,I,L show elasticity maps of fluid DOPC SLBs after the addition of H_2_O_2_ and Cu^2+^ ions. After the addition of H_2_O_2_, distinct decreases of the Young modulus were observed from 63.7 ± 5.2 MPa to 41.5 ± 3.9 MPa for control DOPC SLB (Δ*E* = −22.2 ± 9.1 MPa). The similar, but increased change (Δ*E*= −25.5 ± 9.3 MPa) was observed also after the addition of H_2_O_2_ + Cu^2+^ ions. The shifts in elasticity in the presence of flavonoids after the addition of H_2_O_2_ were significantly suppressed (Δ*E*= −5.1 ± 4.3 MPa, Δ*E*= −6.1 ± 5.7 MPa and Δ*E* = −18.8 ± 9.1 MPa for quercetin, myricetin and myricitrin, respectively). The observed decrease in the Young moduli of SLB confirmed the insertion of exposed flavonoids hosted in MSNs that were released during the incubation time and their protective role during lipid peroxidation. In other words, the softening of the SLBs during lipid peroxidation was significantly reduced. The protective ability of the quercetin seems to be the highest among used flavonoids. The same, but somewhat decreased effect was observed also after the addition of H_2_O_2_ and Cu^2+^ ions.

In conclusion, the present study shows a differential alternating effect of used flavonoids, namely, quercetin, myricetin and myricitrin on the behavior of DOPC model membranes before and after the lipid peroxidation process. Without taking in consideration the flavonoid location in respect to their structure and hydrophobicity, they possess a distinct anti-oxidative capacity in the protection of the model membrane towards damage and disordering of the lipid molecules within the membrane maintaining its elasticity and functionality as best as possible. These results should be taken into consideration in order to understand the lipid–flavonoid interaction and ROS–flavonoid– lipid mutual interactions, however further works are necessary in order to understand the anti-oxidative mechanism on the higher organization level as the cells present.

## 3. Materials and Methods

### 3.1. Materials

Propylamine functionalized silica mesoporous nanoparticles, 200 nm particle size, pore size 4 nm, methoxy poly(ethylene glycol) succinimidyl glutarate (average molecular weight 5000 g/mol, were purchased from Sigma-Aldrich Chemie GmbH (Taufkirchen, Germany) and used as received, 1,2-dioleoyl-sn-glycero-3-phosphocholine (DOPC (18:1), Avanti Polar Lipids Inc., Alabaster, Alabama, USA, >99% purity) was used as received for liposome preparation for AFM and zeta potential measurements. All chemicals were of the highest purity commercially available. Quercetin (Mr = 302.24 g mol^−1^, Sigma-Aldrich Chemie GmbH (Taufkirchen, Germany), myricetin (Mr = 318.23 g mol^−1^, Sigma-Aldrich Chemie GmbH (Taufkirchen, Germany), myricitrin (Mr = 464.38 g mol^−1^, Sigma-Aldrich Chemie GmbH (Taufkirchen, Germany), chloroform (CHCl_3_, Lach-ner, Neratovice, Czech Republik, >99.5% purity), ethanol (Sigma-Aldrich Chemie GmbH (Taufkirchen, Germany), >99% purity), methanol (MeOH, Lach-ner, Neratovice, Czech Republik, >99.5%) and phosphate buffer saline tablets (PBS, pH 7.4, I = 150 mM, containing 137 mM NaCl, 2.7 mM KCl, 1.5 mM KH_2_PO_4_ and 6.5 mM Na_2_HPO_4_, Sigma-Aldrich Chemie GmbH (Taufkirchen, Germany), H_2_O_2_ (Kemika, Zagreb, Croatia) and CuCl_2_ (Sigma-Aldrich Chemie GmbH (Taufkirchen, Germany) were used as received for nanoparticles, film and dispersion preparation for AFM and IR spectroscopy measurements. Spectrum™ Spectra/Por™ 6 Pre-wetted Standard RC Dialysis Tubing MWCO 8kD 7, 5/12 mm, spectra was used to follow the flavonoid release from the MSNs into the liposomes.

### 3.2. Methods

#### 3.2.1. Preparation and Characterization of MSNs

Methoxy-poly(ethylene glycol) succinimidyl glutatare (mPEG-SG) with a molecular weight of 5000 g mol^−1^ was used in all experiments to PEGylate the pre-coated propyl-amine nanoparticles on the surface using the method previously reported with some modifications [62]. Propyl-amine functionalized mesoporous silica nanoparticles 100 mg and 100 mg of methoxy-poly(ethylene glycol) succinimidyl glutarate (m-PEG-SG, MW 5000) were dissolved in 50 mL of ethanol (EtOH). The mixture was stirred for 24 h at 25 °C (50 rpm) to induce the formation of covalent bonds between propyl-amine groups on the surface of the mesoporous SiO2 nanoparticles and succinimidyl groups of PEG. After that, the resulting mixture was removed to an ultrasonic bath to keep the mixture homogeneous. The resulting MSNs were separated from an unreacted mPEG-SCM by five cycles of centrifugation (6000 rpm) and redispersed in ethanol. The dispersion of nanoparticles (MSNs) was left on air overnight to dry.

##### Field Emission Scanning Electron Microscope (FE-SEM)

Field emission scanning electron microscope (FE-SEM) JSM-7000F (JEOL) was used for the observation of particle morphology. The FE-SEM was connected to the Oxford Instruments EDS/INCA 350 energy dispersive X-ray analyzer for elemental analysis. Samples dispersed at an appropriate concentration were cast onto a glass sheet at room temperature and imaged. The size distribution was determined using Image-J (Media Cybernetics Inc., Rockville, USA) by measuring diameters of at least 500 MSNs based on FE-SEM images and presented as a histogram of the nanoparticle diameters.

##### X-ray Powder Diffraction X-ray Diffraction (XRPD)

Here, X-ray diffraction in polycrystalline is used to determine crystalline size, crystal and amorphous material differentiation and to solve and crystallize the crystalline structure. In diffraction structural analysis, monochromatic x-ray radiation with small wavelengths is used in the range from λ = 0.05 to 0.25 nm. Since the λ of X-rays approximates the size of the atoms, this radiation is suitable for determining the structural arrangement of the atoms and molecules of different materials. The position of the diffraction maximum is determined by the crystal grating, the size and form of the unit grid, and the intensity of the diffraction peak atoms of the atom and their spatial deployment in the unit cell according to the requirements of the symmetry, i.e., the crystal structure. In addition to determining the position of the diffraction lines that are directly related to the size and shape of the unit grid, a lot of additional information is obtained from the data that affects the intensity of the individual maximum lines. The structural features of the prepared sample were studied and characterized by powdered X-ray diffraction at room temperature using a Philips MPD 1880 diffractometer with monochromatic CuK a radiation (λ = 0.1541 nm). All samples were recorded at angle 2–15 in the range of 10°–70° with a 0.02° step with a fixed time of 10 s per step.

##### Fourier-Transform Infrared Spectroscopy (FTIR Spectroscopy)

FTIR spectroscopy is a technique used to obtain an infrared spectrum of a solid, liquid or gas. An FTIR spectrometer simultaneously collects high-spectral-resolution data over a wide spectral range. This confers a significant advantage over a dispersive spectrometer, which measures intensity over a narrow range of wavelengths. The term Fourier-transform infrared spectroscopy originates from the fact that a Fourier transform (a mathematical process) is required to convert the raw data into the actual spectrum. FTIR spectra were measured on an ABB Bomem MB102 spectrometer, equipped with CsI optics and a DTGS (deuterated triglycine sulfate) detector. All spectra were collected with a nominal resolution of 4 cm^−1^ and 32 scans at 25 °C. The samples were dried and mixed with KBr to be compressed to a plate for measurement.

##### Atomic Force Microscopy

The MSNs topography and morphology was determined using the MultiMode Scanning Probe Microscope with a Nanoscope IIIa controller (Bruker, Billerica, MA, USA) with SJV-JV-130V (“J” scanner with vertical engagement); vertical engagement (JV) 125 μm scanner (Bruker Instruments, Inc). Of the MSNs dispersion (20 mg/mL) 4 μL was deposited directly onto the freshly cleaved mica (Mica Grade V-4, 9.9 mm disc) mounted on a SPM sample mounting disc. After 60 s the mica surface was washed out with 100 μL Milli-Q water in order to remove unadsorbed MSNs. The washing procedure was repeated two times. The mica surface was dried in air for 3 h. The AFM imaging was performed in the Tapping Mode® under ambient conditions in air using a R-TESPA probe (Bruker, Billerica, MA, USA, Nom. Freq. 300 kHz, Nom. spring constant of 40 N/m). The tapping force, calculated as the ratio of engaged to free amplitude cantilever oscillations (A/A_0_) was maintained at low force (0.8–0.9), which is appropriate for the study of soft and deformable samples. The linear scanning rate was optimized between 1.0 and 1.48 Hz at the scan angle 90°. Imaging and collecting the data was performed with a maximal pixel number (512 × 512). The analysis of images was performed using the offline AFM NanoScopeTM software (Digital Instruments, Santa Barbara, CA, USA, Version V614r1 and V531r1). All images were presented as raw data except for the first-order two-dimensional flattening.

#### 3.2.2. Loading and Release Kinetics of Flavonoids from MSNs

100 mg pegylated MSNs were added to the 15 mL of saturated solution of quercetin, myricetin and myricitrin and were mixed on a stirrer during 24 h at 40 °C. The optimization of the ratio between the weight of added flavonoids and MSNs was performed and the ratio of w(flavonoids):w(MSNs) = 3:1 was found as optimal for the best loading efficiency. After completion, the supernatant was removed and the flavonoid loaded MSNs were washed three times in ethanol (EtOH). The same procedure was repeated (taking 1 mL of DLS suspension, 2 mL of supernatant for DLS, washing with EtOH three times and transferred to a plastic cup to evaporate the EtOH overnight). The analyses were performed after the EtOH was evaporated and the sample was dried.

The significant surface area, high pore volume and pore size of the synthesized MSNs ensured access to their potential to be promising drug delivery vehicles for flavonoids. Loading of flavonoids was performed in pure ethanol. Thirty milligrams of MSNs were added to 30 mL of a flavonoid-saturated solution. The suspension of MSNs was stirred for 24 h to allow for diffusion into the pores. Flavonoid loaded MSNs were separated by applying a centrifuge and dried in a desiccator overnight. Successful loading of flavonoids was confirmed by UV/VIS spectroscopy, zeta potential measurements, Brunauer–Emmet–Teller (BET) analysis and Fourier transform infrared (FTIR) spectroscopy.

##### UV/VIS Spectroscopy

The Beer–Lambert law (or Beer’s law) was applied to determine the amount of flavonoids loaded into MSNs, UV/VIS spectroscopy was employed to directly measure the flavonoid concentration loss in pure EtOH supernatant above synthesized MSNs measuring the absorbance at wavelength λ = 375 nm). Compared with the flavonoid concentration supernatant before adding the synthesized MSNs, the concentration loss was determined using a calibration curve in pure EtOH (Appendix A)). All measurements for quercetin, myricetin and myricitrin were performed at temperature 25 °C and λ = 380 nm, λ = 375 and λ = 380 nm, respectively. The loading efficiency (LE), the ability of the material to entrap a certain active substance was defined as the ration between weight of the loaded flavonoid into MSNs and the weight of the MSNs.

##### Zeta Potential Measurements

The zeta (ζ) potential of MSNs was measured using a Zetasizer Nano ZS (Malvern, UK) equipped with a green laser (532 nm) using the M3-PALS technique. All measurements were conducted at 25 °C. Data processing was done by the Zetasizer software 6.32 (Malvern instruments, Malvern, UK). Results were reported as an average value of three independent measurements. The change in the zeta potential of MSNs before and after their exposure to feeding solution indicated successful flavonoid loading.

##### Brunauer–Emmet–Teller (BET) Analysis for MSNs Porosity Determination

Nitrogen adsorption-desorption measurements were performed on an ASAP2020 (Micromeritics, USA) accelerated surface area analyzer at 77 K. Before measuring, the samples were degassed in a vacuum at 120 °C for at least 6 h.

##### UV/VIS Spectroscopy

According to Kurepa et al. [19], the release kinetics of flavonoids from the MSNs (60 mg) in 30 mL PBS or EtOH/PBS (vol. 50/50) was quantified by UV/VIS absorption measurements (Varian Cary 100 Bio spectrophotometer, 10 mm quartz cuvettes) of the supernatant solution (1.5 mL) during 160 h, 80 h and 175 h for quercetin, myricetin and myricitrin, respectively. Each aliquot of measured supernatant was replaced with the same aliquot (1.5 mL) of fresh PBS or mixture EtOH/PBS (vol. 50/50) for maintaining the volume of the supernatant constant. Temperature in the measuring compartment was controlled and maintained at 25 °C. The calibration curve was drawn by dissolving different amounts of flavonoids in PBS or mixture EtOH/PBS (vol. 50/50) and after filtration of supernatant through filter (F2613-3, PTFE 0.45 µm) measuring the peak maximum in the UV absorption spectra (λ_max_ = 375 nm). The linearity of calibration was found to be valid from 1 × 10^−6^ mol dm^−3^ to 1 × 10^−4^ mol dm^−3^ with correlation coefficients for quercetin all approaching 1.00.

#### 3.2.3. Protective Role of Flavonoids during Lipid Peroxidation Induced by Addition of H_2_O_2_

##### Preparation of DOPC Liposome and Supported Lipid Bilayer (SLB) With and Without Inserted Flavonoids

Pure DOPC liposomes were prepared by preparation of stock solutions of dissolved DOPC in chloroform in order to get the mixed solutions with an adjusted molar ratio of flavonoid in respect to lipids. The procedure has been already reported in our previous papers [63,64]. Shortly, after rotary evaporation of the solvents, the remaining lipid films (pure or mixture of flavonoid and lipid in adjusted ratio) were dried in a vacuum for an hour and then dispersed by gentle manual shaking at 40 °C in 1 mL of phosphate buffer saline (PBS, pH 7.4, *I* = 150 mM, containing 137 mM NaCl, 2.7 mM KCl, 1.5 mM KH_2_PO_4_ and 6.5 mM Na_2_HPO_4_) [63]. During rehydration, the lipid film was gradually scraped off the wall of the glass bottle layer-by-layer and formed cloud-like floaters in the solution. The liposome suspension was left to swell and stabilize overnight at a temperature far away from melting point. The final DOPC concentration in all suspension samples was adjusted to 0.5 mg mL^−1^. The insertion of flavonoids into the liposomes was performed by immersion of flavonoid loaded MSNs placed within a membrane dialysis tube (standard RC Membrane) into 30 mL DOPC liposome dispersion during 24 h. The loss of lipids and subsequent concentration reduction of lipids and flavonoids during the process of extrusion was avoided by using prepared multilamellar vesicles (MLV) dispersions without the process of extrusion.

In order to clarify the effect of the insertion of three distinct flavonoids, quercetin, myricetin and myricitrin, released from MSNs on the topography, organization and nanomechanical properties of DOPC SLB, we performed AFM measurements of the SLBs. To prepare SLB for AFM imaging and force spectroscopy measurements, the drop of MLV liposome suspension was added to fluid cell with a mica plate and kept at 25 °C. After liposome adsorption, the remaining liposomes were removed by washing the surface with PBS, and allowed to be thermostatted at 25 °C. The liposome adsorption during sample deposition and liposome spreading over the mica surface during AFM imaging resulted in the formation of a continuous uniform SLB as a consequence of the sum of the contributions of the electrostatic interactions between liposomes and support liposome alone as well as the surrounding aqueous medium [63]. The lipid peroxidation process has been induced by the addition of 5 µL H_2_O_2_ (10^−5^ M and 5 µL 10^−5^ CuCl_2_) to the liposome dispersion 1 h before the formation of the SLB from treated liposomes suspension. For AFM measurements all supported lipid bilayer (SLB) samples were prepared under the same experimental conditions by the drop deposition method on freshly cleaved mica attached to a metal disc. A volume of 100 µL MLV suspension were pipetted directly onto the mica substrate, incubated for 10 min and flushed with the filtered (0.22 µm Whatman) PBS solution.

##### Atomic Force Microscopy Imaging of SLB in Fluid and Force Spectroscopy Before and After Induced Oxidative Stress

AFM images were obtained by scanning the supported lipid bilayers on the mica surface in fluid using an AFM FastScan Dimension (Bruker Billerica, USA) operated using the new PeakForce QNM mode. Imaging was performed under 25.0 °C for the supported lipid bilayers. The temperature that sensor displays, and the real temperature of the sample were adjusted so there was not the temperature gradient between the displayed and the real temperature. Therefore, the calibration of the temperature individual sample holders was unnecessary. In our AFM experiment in a liquid environment, the mica sample was glued directly to the metallic holder. The whole sample remained attached to the microscope scanner by a magnet. The AFM was allowed to equilibrate thermally before each sample imaging. The AFM measurements were obtained using Scanfastsyst – Fluid + Bruker probes having the spring constant (Nom. *k* = 0.7 Nm^−1^; Nom. resonant freq. ν = 150 kHz). The deflection sensitivity was calibrated, but not the tip radius (the nominal value was used; *R* = 2 nm) in accordance with Atwood [40]. The cantilever was calibrated using the thermal tune method as described by Jazvinšćak Jembrek et al. [28]. AFM images were collected at random spot surface sampling (at least four areas per sample) for each analyzed sample. The quantitative mechanical data was obtained by measuring the Derjaguin–Muller–Toporov model (DMT modulus)/Pa using the Bruker software. All images are presented as raw data except for the first-order two-dimensional flattening. Processing and analysis of raw data were carried out using the NanoScope Analysis software (Version 1.90). To obtain the Young’s Modulus, the retract curve was fit using the Derjaguin–Muller–Toporov model (DMT modulus) [65].

## 4. Conclusions

The obtained results would significantly improve the understanding of interactions between model membranes with flavonoids (free or loaded in MSNs) and will provide an insight into the molecular mechanisms of flavonoid protective activity. Mesoporous silica structures could be considered as universal, and promising drug delivery material that is particularly able to load and release with respectable efficiency flavonoids of different physico-chemical and/or structural properties. Upon therapeutic intervention with flavonoid loaded MSNs, significant membrane protection could be achieved. The AFM analysis revealed that flavonoid suppressed H_2_O_2_-provoked changes in model membrane elasticity and its morphological properties, thus confirming its neuroprotective activity. The obtained results indicate the potential of AFM-measured parameters as a biophysical marker of oxidative stress-induced membrane degeneration. In general, this study suggests that AFM should be used as a highly valuable technique in other biomedical applications aimed at screening and monitoring of drug-induced effects at the membrane level that should be extended to the cellular level.

## Figures and Tables

**Figure 1 ijms-20-02709-f001:**
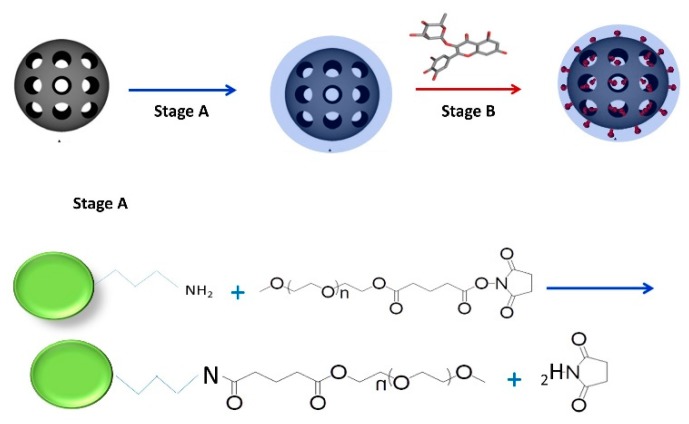
Synthesis of the flavonoid loaded mesoporous silica nanoparticles (MSNs). Stage A: PEGylation of propylamine MSN; Stage B: Loading of flavonoids.

**Figure 2 ijms-20-02709-f002:**
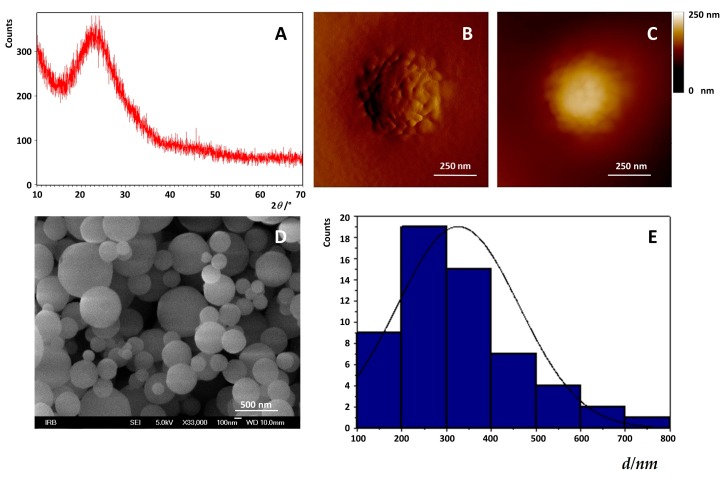
(**A**) The diffraction pattern of the investigated powdered MSNs, (**B**) amplitude and (**C**) 2D height image of MSNs. (**D**) Field emission scanning electron microscope (FE-SEM) micrograph of MSNs as used. (**E**) The histogram of the size distribution of MSN.

**Figure 3 ijms-20-02709-f003:**
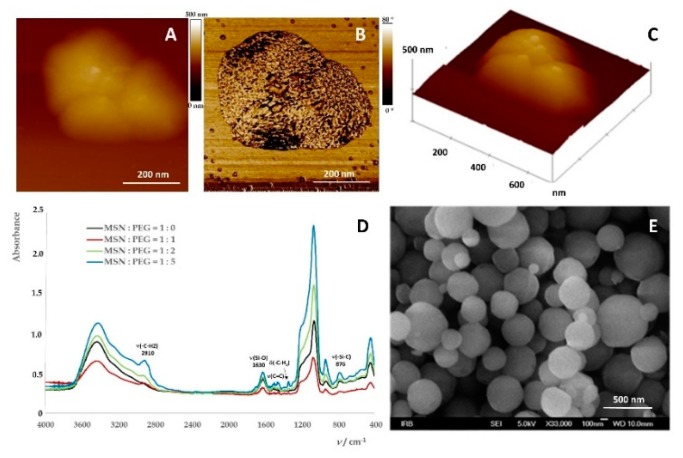
(**A**) Morphology of MSNs coated with PEG. Top view of the 2D atomic force microscopy (AFM) height image, vertical scale 500 nm, (**B**) phase image, vertical scale 80°, indicating PEG coatings on MSNs (white color on the MSNs), (**C**) 3D height image of an aggregate consisting of three MSNs, (**D**) FTIR spectra of pure MSNs (black); MSNs with PEG coating in weight ratio 1:1( red); 1:2 (green); 1:5 (blue), (**E**) FE-SEM micrograph of the PEG coated MSNs.

**Figure 4 ijms-20-02709-f004:**
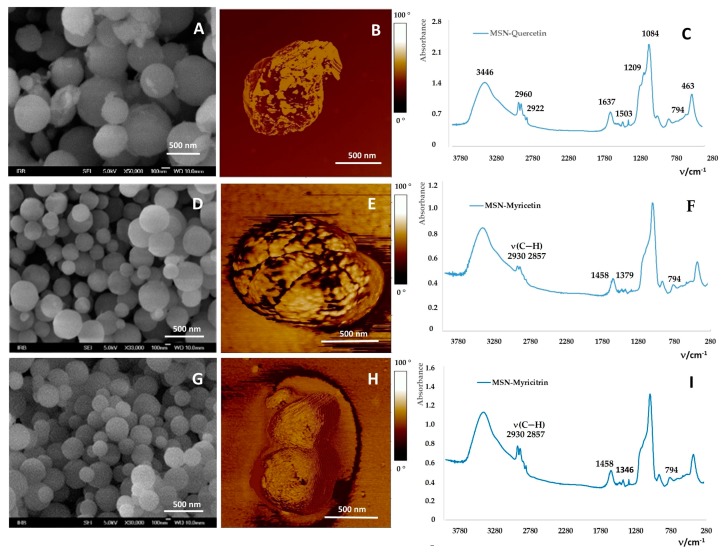
(**A**–**C**) Characterization of flavonoid loaded MSNs: With quercetin, (**D**–**F**) myricetin and (**G**–**I**) myricitrin. (**A**, **D**, **G**) FE-SEM of flavonoid loaded MSNs, (**B**, **E**, **H**) AFM phase images and (**C**, **F**, **I**) FTIR spectra of flavonoid loaded MSNs.

**Figure 5 ijms-20-02709-f005:**
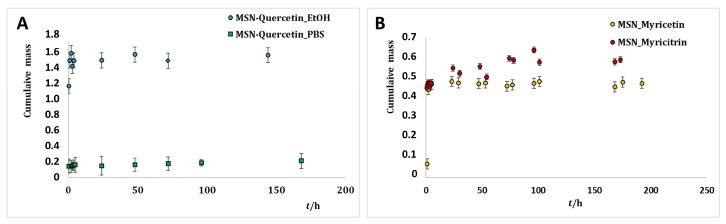
(**A**) Cumulative release profiles of quercetin in EtOH/H_2_O and PBS, (**B**), myricetin and myricitrin in EtOH/H_2_O.

**Figure 6 ijms-20-02709-f006:**
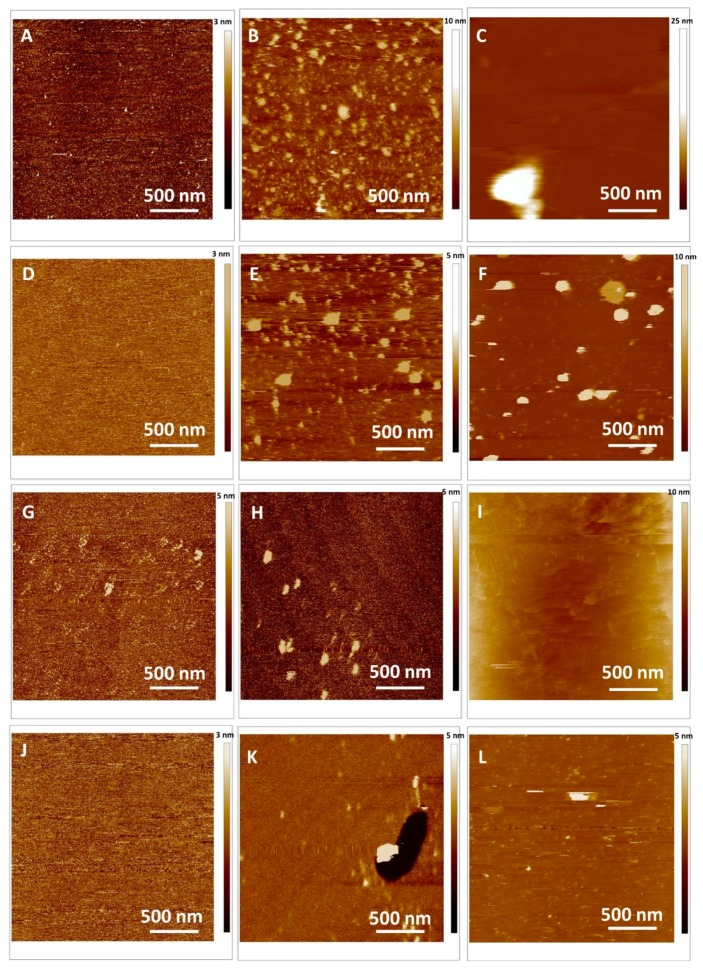
Top view of 2D height AFM images on the model (DOPC) SLB: (**A**–**C**) Control, (**D**–**F**) quercetin loaded DOPC; (**G**–**I**) myricetin loaded DOPC and (**J**–**L**) myricitrin loaded DOPC. The lipid peroxidation induced by (**B**, **E**, **H**, **K**) H_2_O_2_, (**C**, **F**, **I**, **L**) H_2_O_2_ and Cu^2+^.

**Figure 7 ijms-20-02709-f007:**
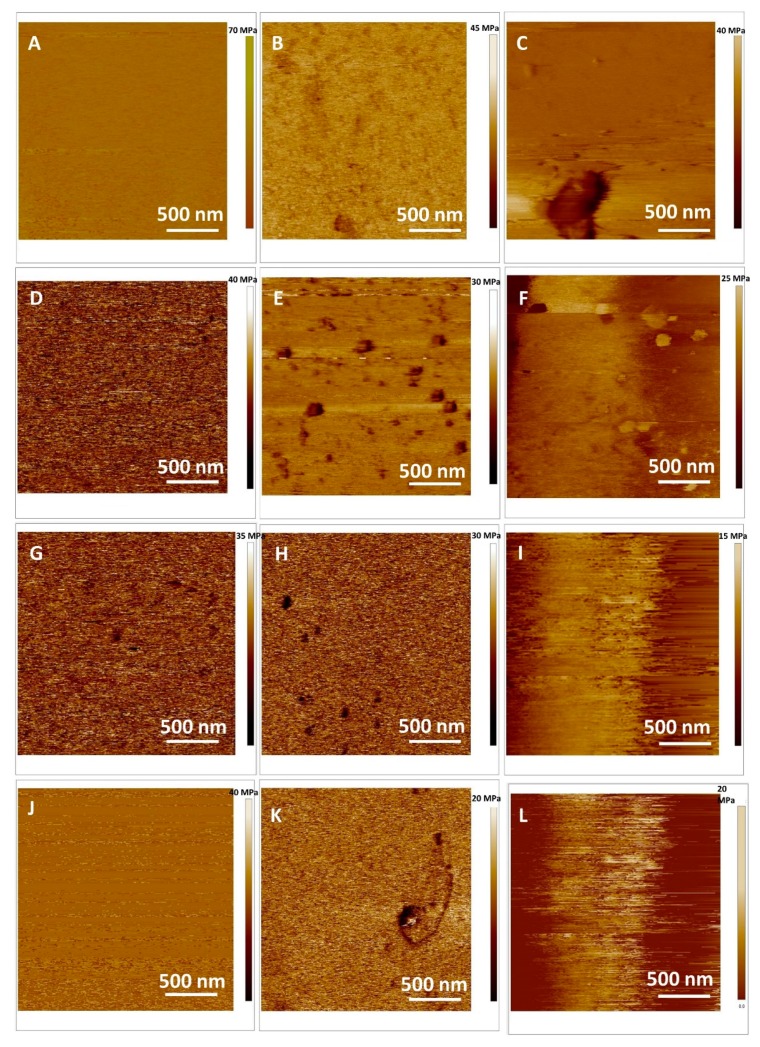
Young modulus maps of the model lipid (DOPC) SLB: (**A**–**C**) Control; (**D**–**F**) quercetin loaded DOPC, (**G**–**I**) myricetin loaded DOPC and (**J**–**L**) myricitrin loaded DOPC. The lipid peroxidation induced by (**B**, **E**, **H**, **K**) H_2_O_2_, (**C**, **F**, **I**, **L**) H_2_O_2_ and Cu^2+^.

**Table 1 ijms-20-02709-t001:** The porosity, morphology and stability of MSNs in powder and dispersed in water.

Powder MSN	
Specific surface/m^2^ g^−1^	693.78
Pore volume/cm^3^ g^−1^	0.84
Pore size/nm	4.82
Zeta potential/mV	+26 ± 2
*d*_H_/nm	913 ± 180
^1^*d*/nm	326 ± 137

^1^ FE-SEM (*n* = 57).

**Table 2 ijms-20-02709-t002:** The zeta potential of MSNs and the average diameter size of aggregates formed by dispersing of MSNs in water.

MSN_PEG_5000_	
Zeta potential/mV	+27 ± 1
*d*_H_/nm	932 ± 91

The rest of the MSNs were used further for the loading of flavonoids.

**Table 3 ijms-20-02709-t003:** Specific surface area, pore volume, pore size and loading efficiency (LE) for flavonoid loaded MSNs determined by BET analysis and UV/VIS spectroscopy.

	Quercetin	Myricitrin	Myricetin
Specific surface area/m^2^ g^−1^	544.58	546.01	562.71
Pore volume/cm^3^ g^−1^	0.6404	0.6527	0.73
Pore size/nm LE (%)	4.70 27 ± 9 (*n* = 6)	4.78 8.6 ± 0.6 (*n* = 3)	3.12 4 ± 2 (*n* = 5)

**Table 4 ijms-20-02709-t004:** Zeta potential values of DOPC liposomes (*γ* = 0.5 mg mL^−1^) with and without inserted flavonoids at 25 °C before and after induced lipid peroxidation process by the addition of H_2_O_2._

Sample	DOPC	DOPC/Quercetin Loaded MSNs	DOPC/Myricetin Loaded MSNs	DOPC/Myricitrin Loaded MSNs
^1^ζ/mV	−6.1 ± 1.1	−14.4 ± 4.7	−6.7 ± 1.3	−11.8 ± 3.2
^2^ζ/mV	−17.8 ± 6	−8.1 ± 1.4	−10.4 ± 2.2	−6.3 ± 2

^1^ Before addition of H_2_O_2_, ^2^ after addition of H_2_O_2_.

**Table 5 ijms-20-02709-t005:** The effect of induced lipid peroxidation on the roughness (*R*_a_), bilayer thickness (*d*) and the Young Modulus (*E*) of the model DOPC SLB (*n* = 6).

Sample	*R*_a_/nm	Δ*R*_a_/nm	*E*/MPa	Δ*E*/MPa	*d*/nm
Control DOPC	0.08 ± 0.01		63.7 ± 5.2		7.1 ± 0.3 (*n* = 256)
Control DOPC/H_2_O_2_	0.33 ± 0.05	+0.25 ± 0.06	41.5 ± 3.9	−22.2 ± 9.1	6.9 ± 0.2 (*n* = 201)
Control DOPC/H_2_O_2_ + Cu^2+^	0.84 ± 0.02	+0.76 ± 0.01	38.2 ± 4.1	−25.5 ± 9.3	6.7 ± 0.5 (*n* = 322)
DOPC/Quercetin	0.11 ± 0.05		40.6 ± 2.7		7.2 ± 0.4 (*n* = 276)
DOPC/Quercetin/H_2_O_2_	0.18 ± 0.06	+0.07 ± 0.11	35.5 ± 1.6	−5.1 ± 4.3	7.0 ± 0.3 (*n* = 151)
DOPC/Quercetin/H_2_O_2_+ Cu^2+^	0.81 ± 0.17	+0.70 ± 0.22	16.6 ± 7.4	−9.2 ± 9.0	6.8 ± 0.5 (*n* = 123)
DOPC/Myricetin	0.12 ± 0.06		31.4 ± 2.9		7.2 ± 0.6 (*n* = 99)
DOPC/Myricetin/H_2_O_2_	0.22 ± 0.02	+0.10 ± 0.08	25.3 ± 2.8	−6.1 ± 5.7	7.0 ± 0.2 (*n* = 143)
DOPC/Myricetin/H_2_O_2_+ Cu^2+^	1.02 ± 0.05	+0.80 ± 0.07	17.6 ± 2.1	−13.8 ± 5.0	6.9 ± 0.5 (*n* = 101)
DOPC/Myricitrin	0.19 ± 0.09		37.6 ± 4.8		7.3 ± 0.2 (*n* = 175)
DOPC/Myricitrin/H_2_O_2_	0.30 ± 0.04	+0.11 ± 0.13	18.8 ± 4.3	−18.8 ± 9.1	7.1 ± 0.3 (*n* = 198)
DOPC/Myricitrin/H_2_O_2_+ Cu^2+^	0.81 ± 0.04	+0.62 ± 0.13	14.7 ± 3.4	−22.9 ± 8.2	7.0 ± 0.4 (*n* = 125)

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
