# Peer review of "Enhanced Protection of Biological Membranes during Lipid Peroxidation: Study of the Interactions between Flavonoid Loaded Mesoporous Silica Nanoparticles and Model Cell Membranes"

_ijms, 2019, doi:10.3390/ijms20112709_

Round 1
Reviewer 1 Report
Results and Discussion is mixed with Materials and Methods. Separate them clearly.
Materials and Methods should contain citations. In Materials and Methods do not repeat methods in each step.
There is no real meaning of X-ray powder diffraction data in this paper (Fig 2 A)
Table 1 is incorrect.
Fig 3: (A) is probably amplitude (error) image. (B) there is no scale bar for the phase image. (C) If you have a spherical particle in 250-500 nm diameter, height of the particle should be 250-500 nm.
Row 206: Phase image is Fig 3B.
Fig 4: (A) Scale bar is incorrect, magnification is different . (I) Why do you have an extra spectrum?
Row 327. Your paper is about 3 flavonoids.
Row 330. Please specify "membrane bag" (By the way, this part should be in Mat Meth)
Row 336. How do you confirm the presence of flavonoid in the membrane by AFM. Solve the liposomes, and measure the flavonoid content of them by UV-VIS described in this paper.
Row 395. Based on Fig S1, I am not sure, that you have single supported layer. Probably yes, but these data are not real evidence. You prove it based on force curves, or measure the deep of holes.
Fig 6: Based on 3D images, (B) and (C) are confused (see S2). It seems, (I) has a flattening error.
Row 413. There is no log P value for myricitin.
Fig. 7: Based on Fig 6. (I) and (L) are confused. Did the same areas were measured on Fig 6 and Fig 7, respectively?
Table 5. Young's modulus data are out of the range of Fig 7.
Mat Meth: use "Sigma Aldrich" name identically.
Row 510. Quercetin
Row 515. What about FeCl2? CuCl2 was used in this research.
Row 696. The only citation in Mat Meth is in different format.
Author Response
Reviewer 1.
Comments and Suggestions for Authors
Results and Discussion is mixed with Materials and Methods. Separate them clearly.
Answer: As suggested, we made corrections in the manuscript in order to separate Results and Discussion from Materials and Methods.
Materials and Methods should contain citations. In Materials and Methods do not repeat methods in each step.
Answer: As both reviewers pointed out, we rewrote Materials and Methods. As suggested, we added reference to Materials and Methods.
There is no real meaning of X-ray powder diffraction data in this paper (Fig 2 A)
Answer: XRD pattern which does not feature any diffraction lines can not, of course, be used for the qualitative identification however that was not an intention. The idea was to show that materiel is, indeed, amorphous.
Table 1 is incorrect.
Answer: Table 1. has been corrected.
Fig 3: (A) is probably amplitude (error) image. (B) there is no scale bar for the phase image. (C) If you have a spherical particle in 250-500 nm diameter, height of the particle should be 250-500 nm.
Answer: According to the request of the reviewer, the amplitude (error) image has been replaced with height image (A); the scale has been inserted on scale bar (B); the 3D_height image has been corrected for the vertical scale (C).
Row 206: Phase image is Fig 3B.
Answer: Fig 3C has been corrected to Fig 3B.
Fig 4: (A) Scale bar is incorrect, magnification is different. (I) Why do you have an extra spectrum?
Answer: Scale bar on Fig 4(A) has been corrected. The extra spectrum in Fig (I) has been removed.
Row 327. Your paper is about 3 flavonoids.
Answer: The mention of two flavonoids has been corrected into three distinct flavonoids, quercetin, myricetin and myricitrin.
Row 330. Please specify "membrane bag" (By the way, this part should be in Mat Meth)
Answer: The "membrane bag" has been applied for molecular separation of released flavonoids from MSNs and liposome dispersion. It has been corrected to membrane dialysis tube and removed and specified in Mat Meth.
Row 336. How do you confirm the presence of flavonoid in the membrane by AFM. Solve the liposomes, and measure the flavonoid content of them by UV-VIS described in this paper.
The presence of the flavonoids in the membrane has been confirmed using zeta potential measurements. Additionally, it is also confirmed by AFM force spectroscopy measurement revealing both the increase in the bilayer thickness and the decrease in the Young modulus value of the SLBs. Since the all samples are prepared and measured at the strict same conditions, the only difference in the mentioned values have been ascribed to the inclusion of flavonoids into the membrane.
We tried to extract the lipids into hydrophobic phase and found out the content of the hydrophobic phase (chloroform). But, due to the presence of the flavonoids (diferent hydrophobicity) in the chloroform, the aggregation process appeared. This aggregates disabled the obtaining of the UV/VIS specta good quality. We will work on this study in the future to get quantitative approach and results regarding not only the composition and molar ratio of the lipids and flavonoids in the liposomes, but also regarding the composition and ratio of the products of the lipid peroxidation. In the present phase of our study, the quantitative approach was no intention, but the enhanced and prolonged protection of the membrane by released flavonoids from the MSNs.
Row 395. Based on Fig S1, I am not sure, that you have single supported layer. Probably yes, but these data are not real evidence. You prove it based on force curves, or measure the deep of holes.
Answer: The protocol for the SLB formation is described in our recent studies (http://dx.doi.org/10.1016/j.chemphyslip.2014.11.001 and https://pubs.acs.org/doi/abs/10.1021/acs.jpcb.5b00898 ) as well as the bilayer thickness determination. However, in nanomechanical measurements beside the elasticity maps, we also measured the force curves to confirm the formation of the single supported lipid bilayer and found out its thickness. The obtained values of SLB thickness are more accurate than the height measurements obtained using cross section profiles near the SLB defects. The results have been summarized in Table 5 and FigureS3.
Fig 6: Based on 3D images, (B) and (C) are confused (see S2). It seems, (I) has a flattening error.
Answer: The Figures (B) and (C) have been corrected. The order of the Figures (B) and (C) was replaced in S2 and now is correct. Figure (I) has wrong flattening procedure which is now correct and replaced.
Row 413. There is no log P value for myricitin.
Answer: The log P = 0.25 for myricitrin has been inserted in the manuscript.
Fig. 7: Based on Fig 6. (I) and (L) are confused. Did the same areas were measured on Fig 6 and Fig 7, respectively?
Answer: All samples have been measured at the same conditions and the same imaging areas of 2×2 mm2 have been measured.
Table 5. Young's modulus data are out of the range of Fig 7.
Answer: The Young's modulus scales have been corrected in Fig. 7
Mat Meth: use "Sigma Aldrich" name identically.
Answer: The name Sigma-Aldrich has been uniformed in the manuscript.
Row 510. Quercetin
Answer: The name has been corrected.
Row 515. What about FeCl2? CuCl2 was used in this research.
Answer: FeCl2 has been replaced with CuCl2.
Row 696. The only citation in Mat Meth is in different format.
Answer: The citation has been corrected.
Reviewer 2 Report
The work describes the use of biodegradable mesoporous silica nanoparticles to overcome problems related with stability and solubility for flavnoid antioxidants. quercetin, myricetin and myricetrin have been investigated to see their protective effect to a model cell membrane and structure activity relationship was also determined. The authors also tried to support their findings utilizing various methodologies and instrumentations. Below are some my observations and comments to improve the article if accepted to be published after revision.
1. Despite the detailes given in the introduction it looks to lack some flow and I recommened the authers to rewrite spacially the third paragraph needs some polish. Toward the end of the third paragraph, they have mentioned two references in a different way.
2. The quality of the figures should be improved and the same format and scale have to be used specially the FT-IR section.
3. Figure one should be self explanatory and I expect the authers to show the chemistries involved in coupling the PEG to the silica and the conjugate coupling to the flavnoids.
4. In addition to the low figure quality and scale usage discrepancies the FT-IR is not confirmatory to the coupling the flavnoids to the nanoparticle PEG conjugate. The regions in the FT-IR the authors claim to indicate covalent coupling are also seen in the unconjugated nanoparticles. I recommend to the authors so try conjugating the PEG with the flavnoids in solution the couple with the amine functionalized silica nanoparticles.
5. The authors should put all the methods they used in the materials and methods section. Some of the methods are mentioned in the result and discussion section. I found it to reduce the quality.
Author Response
The work describes the use of biodegradable mesoporous silica nanoparticles to overcome problems related with stability and solubility for flavonoid antioxidants. quercetin, myricetin and myricetrin have been investigated to see their protective effect to a model cell membrane and structure activity relationship was also determined. The authors also tried to support their findings utilizing various methodologies and instrumentations. Below are some my observations and comments to improve the article if accepted to be published after revision.
1. Despite the details given in the introduction it looks to lack some flow and I recommended the authors to rewrite specially the third paragraph needs some polish. Toward the end of the third paragraph, they have mentioned two references in a different way.
Answer: As reviewer pointed out, we rewrote the Introduction, specially the third paragraph. As suggested, we corrected references.
2. The quality of the figures should be improved and the same format and scale have to be used specially the FT-IR section.
Answer: According to request, the quality of the FT-IR figures has been improved regarding the format and scale. AFM images have been also improved.
3. Figure one should be self explanatory and I expect the authors to show the chemistries involved in coupling the PEG to the silica and the conjugate coupling to the flavonoids.
Answer: As requested, the chemistry behind the coupling the PEG to the silica MSNs has been shown in the manuscript and Figure 1.
4. In addition to the low figure quality and scale usage discrepancies the FT-IR is not confirmatory to the coupling the flavonoids to the nanoparticle PEG conjugate. The regions in the FT-IR the authors claim to indicate covalent coupling are also seen in the unconjugated nanoparticles. I recommend to the authors so try conjugating the PEG with the flavonoids in solution the couple with the amine functionalized silica nanoparticles.
Answer: As reviewer pointed out, we rewrote and explained in manuscript the FTIR results. The present system consists of MSNs covalent coupled with PEG coating as their stabilizer. The flavonoids did not covalent coupled with PEG. Instead, flavonoids are coupled with the surface of the MSNs by hydrogen bond described and specified in manuscript. Besides, we tried to load the flavonoids to MSNs and then to make the coupling with PEG. Unfortunately, both the stabilization of the MSNs and the loading efficiency did not reach an adequate level. A detailed analysis of the spectra (no scope of this study) confirms the presence and structural integrity of flavonoid molecules within the silica pores. Moreover, changes in the position of flavonoid vibrational modes between 1400 and 1600 cm-1, more precisely 1503 cm-1, 1458 and 1379 cm-1 and 1349w cm-1 and 1461w cm-1 for quercetin, myricetin and myricitrin loaded MSNs, respectively, i.e. (nC=O, aromatic n(C=C) and n(C-O/dO–H) combination modes, indicate hydrogen bonding interactions from the silica surface groups to the carbonyl of flavonoids [10.1016/j.ejpb.2014.11.022]. The hydrogen bond interaction changes most of the substantial vibrations of flavonoids. For example, the d(5-OH) band cantered at 1520 cm-1 is not present in the spectrum of mirycitrin incorporated into MSNs, which indicates the involvement of these functional group in the type of interaction, e.g., in a hydrogen bond with silica surface. For quercetin loaded MSNs, band at 1153 cm-1 was attributable to the C–O stretching in the aryl ether ring and the C–O stretching in phenol, respectively [53]. Bands at 2930sh, 2857 cm-1 were due to CH stretching of PEG. The band at 1379 cm-1 belongs to myricetin (Figure 4F), while in spectrum of myricitrin loaded MSNs are present bands 2930, 1507w, 1461w and 1346w cm-1. (Figure 4I). Band at 1346 cm-1 belongs present myricitrin.
5. The authors should put all the methods they used in the materials and methods section. Some of the methods are mentioned in the result and discussion section. I found it to reduce the quality.
Answer: As both reviewers pointed out, we rewrote Materials and Methods. As suggested, we added reference to Materials and Methods.
Round 2
Reviewer 1 Report
One minor suggestion. Fig 4. A, D, G: magnification of the SEM images are different (x50000, x33000, x30000). Please change the scale bar. Other modifications are accepted.